# Myoglobin Concentration and Oxygen Stores in Different Functional Muscle Groups from Three Small Cetacean Species

**DOI:** 10.3390/ani11020451

**Published:** 2021-02-09

**Authors:** Marina Arregui, Emily M. Singleton, Pedro Saavedra, D. Ann Pabst, Michael J. Moore, Eva Sierra, Miguel A. Rivero, Nakita Câmara, Misty Niemeyer, Andreas Fahlman, William A. McLellan, Yara Bernaldo de Quirós

**Affiliations:** 1Atlantic Center for Cetacean Research, Institute of Animal Health and Food Safety (IUSA), Veterinary School, University of Las Palmas de Gran Canaria, C/Transmontaña s/n, 35413 Las Palmas, Spain; marina.arregui@ulpgc.es (M.A.); miguel.rivero@ulpgc.es (M.A.R.); kita_camara@hotmail.com (N.C.); yara.bernaldo@ulpgc.es (Y.B.d.Q.); 2Department of Biology and Marine Biology, University of North Carolina Wilmington, 601 S. College Road, Wilmington, NC 28403, USA; emilymsingleton@gmail.com (E.M.S.); pabsta@uncw.edu (D.A.P.); mclellanw@uncw.edu (W.A.M.); 3Department of Mathematics, Campus de Tafira s/n, University of Las Palmas de Gran Canaria, 35017 Las Palmas, Spain; pedro.saavedra@ulpgc.es; 4Biology Department, Woods Hole Oceanographic Institution, Woods Hole, MA 02543, USA; mmoore@whoi.edu; 5International Fund for Animal Welfare, Yarmouth Port, MA 02675, USA; mniemeyer@ifaw.org; 6Global Diving Research Inc., Ottawa, ON K2J 5E8, Canada; afahlman@oceanografic.org; 7Fundación Oceanogràphic, Department of Research, Ciutat de les Arts i de les Ciències, Carrer d’Eduardo Primo Yúfera, 1B, 46013 Valencia, Spain; 8Department of Life Sciences, Texas A&M University-Corpus Christi, 6300 Ocean Drive, Unit 5892, Corpus Christi, TX 78412, USA

**Keywords:** *D. delphis*, *S. coeruleoalba*, *S. frontalis*, muscle mass, heterogeneity, aerobic dive limit

## Abstract

**Simple Summary:**

Marine mammals display several physiological adaptations to their marine environment. Higher myoglobin concentrations in their muscles compared to terrestrial mammals allow them to increase their onboard oxygen stores, enhancing the time available to dive. Most previous studies have calculated cetaceans’ onboard oxygen stores by assuming the myoglobin concentration of a single muscle to be representative of all the muscles in the body. In this study, we analyzed this assumption by comparing it to a more precise method that weighs all body muscles and measures myoglobin concentration in different functional groups.

**Abstract:**

Compared with terrestrial mammals, marine mammals possess increased muscle myoglobin concentrations (Mb concentration, g Mb · 100g^−1^ muscle), enhancing their onboard oxygen (O_2_) stores and their aerobic dive limit. Although myoglobin is not homogeneously distributed, cetacean muscle O_2_ stores have been often determined by measuring Mb concentration from a single muscle sample (*longissimus dorsi*) and multiplying that value by the animal’s locomotor muscle or total muscle mass. This study serves to determine the accuracy of previous cetacean muscle O_2_ stores calculations. For that, body muscles from three delphinid species: *Delphinus delphis*, *Stenella coeruleoalba*, and *Stenella frontalis*, were dissected and weighed. Mb concentration was calculated from six muscles/muscle groups (epaxial, hypaxial and *rectus abdominis*; *mastohumeralis*; *sternohyoideus*; and *dorsal scalenus*), each representative of different functional groups (locomotion powering swimming, pectoral fin movement, feeding and respiration, respectively). Results demonstrated that the Mb concentration was heterogeneously distributed, being significantly higher in locomotor muscles. Locomotor muscles were the major contributors to total muscle O_2_ stores (mean 92.8%) due to their high Mb concentration and large muscle masses. Compared to this method, previous studies assuming homogenous Mb concentration distribution likely underestimated total muscle O_2_ stores by 10% when only considering locomotor muscles and overestimated them by 13% when total muscle mass was considered.

## 1. Introduction

Marine mammals store oxygen (O_2_) within the respiratory system, blood and muscle, e.g., [1,2]. The aerobic dive limit (ADL) is defined as the dive duration that can be performed aerobically, beyond which blood lactate levels begin to rise above resting levels [3,4,5,6]. Marine mammals display adaptations that increase the amount of O_2_ stored within these three compartments and conserve those stores to increase the ADL, e.g., [7,8,9,10].

Myoglobin (Mb) is an O_2_ binding protein whose function is to store O_2_ within the muscle and facilitate its diffusion within muscle cells [11]. Myoglobin concentration (Mb concentration) is higher in marine mammal muscle than those of similarly sized terrestrial mammals [2,12]. In marine mammals, the amount of muscle O_2_ has traditionally been calculated by measuring the Mb concentration (g Mb · 100g^−1^ muscle) at a single location within a primary locomotor muscle and assuming a homogeneous distribution of Mb across all the body’s muscles. Most previous studies have calculated total Mb from a single biopsy or muscle sample extrapolated to the locomotor muscle mass (epaxial group, hypaxial group and *rectus abdominis*) or the entire muscle mass, e.g., [13,14,15,16,17]. Nevertheless, several studies have shown significant differences in Mb concentration between muscles [18,19,20,21] and even within a single muscle [22,23,24] in marine mammals.

An accurate determination of O_2_ stored in the muscles is crucial for a proper determination of ADL. The goal of this study was to investigate the effect of the heterogeneity of skeletal muscle Mb concentration on calculations of locomotor muscle, and total muscle, oxygen stores in three species of similar size, closely related pelagic delphinids [25,26]: *Delphinus delphis* (common dolphin), *Stenella coeruleoalba* (striped dolphin), and *Stenella frontalis* (spotted dolphin). Kroeger and colleagues recently investigated the morphological features of a primary locomotor muscle, the *longissimus dorsi*, of these three species, including muscle fiber-type and size, index of mitochondrial volume density, and Mb concentration [27]. Thus, these species are well-suited for this broader investigation of Mb concentration and oxygen storage across muscle groups. For this study, muscles from four distinct functional groups were utilized, including locomotion (epaxial, hypaxial, and *rectus abdominis*), pectoral fin movement (*mastohumeralis*), feeding (*sternohyoideus*), and respiration (*dorsal scalenus*). The Mb concentration values from each of these representative muscles were then used to compare multiple methods for calculating onboard oxygen stores.

## 2. Materials and Methods

### 2.1. Specimens and Collection of Muscle Tissue

Muscle samples were collected from stranded cetaceans on Cape Cod, Massachusetts (USA) from September 2013 to September 2014, and on the Canary Islands (Spain) from October 2014 to March 2017. Eleven individuals from three species were used: *Delphinus delphis* (n = 4), *Stenella coeruleoalba* (n = 3), and *Stenella frontalis* (n = 4) (Table 1). Carcasses were fresh (Code 2: no bloating nor changes in coloration, eatable meat) or moderately fresh (early Code 3: may present some incipient signs of autolysis such as small changes in coloration) [28]. Required permission for the management of stranded cetaceans was issued by the Greater Atlantic Fisheries Regional Office belonging to the National Marine Fisheries Services (NMFS) within the National Oceanic and Atmospheric Administration (NOAA) and the Environmental Department of the Canary Islands’ Government and the Spanish Ministry of Environment. No experiments were performed on live animals. Samples were analyzed within the countries where they were collected; thus, a CITES permit was not needed.

During necropsy, all body muscles, excluding those of the head and the *cutaneous trunci* muscle, which was not separated from the blubber, were dissected based upon the methods of McLellan et al. [29]. They were then weighed and classified by function: locomotion for swimming (upstroke and downstroke of tail fluke), pectoral fin movement, suction feeding, and respiration (Table 2, Figure 1) [30,31,32,33]. Thus, this grouping identifies both “locomotor muscle” and “non-locomotor muscle.” The sum of all muscles in Table 2 was used for total muscle mass.

At least one representative muscle of each functional group was sampled for Mb concentration determination: the *mastohumeralis* muscle (pectoral fin movement), the *sternohyoideus* muscle (suction feeding), and the *dorsal scalenus* muscle (respiration). In the case of locomotor muscles powering swimming, several muscles were sampled given that more and larger muscles are involved: the *longissimus dorsi* in the epaxial muscle group (locomotor upstroke), the hypaxial muscle group (locomotor downstroke), and the *rectus abdominis* muscle (assistance in the tail fluke downstroke) (Table 2). Within the epaxial group, the muscle *semispinalis* does not contribute to the upstroke per se but to the head movement [30]. However, due to the difficulties of separating this muscle from the rest of the epaxial muscle, it was included within the epaxial muscles.

This muscle grouping scheme was used to recognize that cetacean muscles can perform multiple functions as in all mammals. For example, the primary downstroke muscles also contribute to exhalation, and the *sternohyoideus* to inhalation, during a respiratory event [31].

The longest muscles were also sampled at multiple locations: within the epaxial group, the *longissimus dorsi* at the level of the axilla, middle (mid-distance between axilla and anus), and anus; the hypaxial group at the middle and anus levels. The rest of the muscles were sampled at the mid-belly. Muscle samples were wrapped in plastic wrap, placed in sealed bags and frozen at −20 °C until analysis.

### 2.2. Mb Concentration Determination

Muscle Mb concentration was measured in two laboratories: UNC Wilmington in North Carolina, USA, and the University of Las Palmas de Gran Canaria in the Canary Islands, Spain, using methods adapted from Reynafarje by Noren and Williams and Etnier et al. [20,34,35]. Frozen muscle samples of ~0.5 g were thawed, minced and cleaned of apparent fat and connective tissue. Samples were diluted in ice-cold 0.04M phosphate buffer (pH 6.6) to obtain a final dilution of 39.25 mL buffer per gram of tissue. The solution was homogenized (IKA Ultra-Turrax T18) and then centrifuged at 28,000x *g* for 50 min at 4 °C (Beckman J2-MI Centrifuge). Five ml of supernatant were transferred to a test tube and bubbled with carbon monoxide (CO N37 S10, Air Liquide, Spain) at room temperature for 8 min. Approximately 0.02 g of sodium dithionite were added to the sample to ensure a complete reduction of Mb. The sample was bubbled again for 2 min with CO. The absorbance was measured at 538 and 568 nm (Genesys 10 S UV, Thermo Scientific, Thermo Electron Scientific Instruments LLC, Madison, WI, USA). Three muscle samples were analyzed for each sampling location, and three spectrophotometric measurements were obtained for each of the replicates from an individual.

To compare our results with the literature, Mb concentration (g Mb · 100g^−1^ muscle) was calculated using the equation developed by Reynafarje [35], where Mb concentration is determined as the difference in absorbance at two wavelengths.

Mb concentration values for one animal (CET 834) were also calculated using a calibration curve to test if calculated Mb concentration values may vary when using Reynafarje’s equation with different spectrophotometers. Horse Mb standard (Equine skeletal muscle, M1882A, Sigma-Aldrich, Spain) was used to construct the calibration curve (Appendix A). Values obtained with the calibration curve (mg Mb · ml^−1^ solution) were divided by 0.025 g muscle· ml^−1^ (0.5 g muscle were dissolved in 20 mL of solvent (buffer volume + % water in muscle) and then transformed to obtain g Mb · 100g^-1^ muscle. Mb concentration values of CET 834, calculated following Reynafarje (1963) and using the calibration curve, were compared to test both approaches.

### 2.3. Muscle O_2_ Storage Determination

Total Mb within each muscle was calculated to determine locomotor and total muscle O_2_ storage. Since the goal of this study was to investigate how Mb concentration heterogeneity affects calculations of locomotor muscle and total muscle oxygen stores, total muscle myoglobin was calculated in three different ways:(1)Homogeneous myoglobin distribution was assumed across the locomotor muscles. Mb concentration measured at the middle epaxial (*Longissimus dorsi*) muscle position was multiplied by the animal’s total locomotor muscle mass (LMM; see Table 2) (Equation (1)). This method will be referred to as the “locomotor muscle method.”
(1)$$T\mathrm{otal}\;\mathrm{Mb}\;=\;{\left[\mathrm{Mb}\right]}_{\mathrm{Epaxial}\;\mathrm{middle}}\;\cdot \;\mathrm{LMM}$$(2)Homogeneous myoglobin distribution was assumed across all the skeletal muscles. Mb concentration at the middle epaxial muscle was multiplied by the animal’s total muscle mass (TMM) (Equation (2)). This method will be referred to as the “total muscle method.”
(2)$$\mathrm{Total}\;\mathrm{Mb}\;={\left[\mathrm{Mb}\right]}_{\mathrm{Epaxial}\;\mathrm{middle}}\cdot \;\mathrm{TMM}$$(3)Heterogeneous myoglobin distribution was assumed across the skeletal muscles. The Mb for each functional group was calculated by multiplying the Mb concentration of the representative muscle by the mass of all muscles in each functional muscle group (FMG) following Equation (3). In those muscles where Mb concentration was determined in different locations, the mean Mb concentration among or between locations was calculated. This method will be referred to as the “functional muscle group method.”
(3)$$\mathrm{Functional}\;\mathrm{group}\;\mathrm{Mb}\;={\left[\mathrm{Mb}\right]}_{\mathrm{Representative}\;\mathrm{muscle}}\times \mathrm{FMG}$$

Total muscle Mb was then calculated as the summation of Mb values within the different functional muscle groups (Equation (4)).
(4)$$\mathrm{Total}\;\mathrm{Mb}\;=\;\sum^{\;}\mathrm{Functional}\;\mathrm{group}\;\mathrm{Mb}$$

Total O_2_ stored within the muscle was obtained by multiplying the total amount of Mb, calculated by the three methods, by the O_2_ binding capacity of Mb (O_2_Mb: 1.34 mL O_2_ · g^−1^ Mb, assuming complete Mb saturation at the beginning of the dive) (Equation (5)) [36,37].
(5)$${\mathrm{Total}\;O}_{2}\;\mathrm{muscle}\;=\;\mathrm{Total}\;\mathrm{Mb}\;\times O_{2}\mathrm{Mb}$$

### 2.4. Statistical Analysis

#### 2.4.1. Study of Muscle Contribution to Total Mb

Whether the Mb concentration varied between species and muscle groups was assessed using a mixed model [38] and the restricted maximum likelihood method (REML) to fit the variables. The significance of each variable was assessed using the corresponding Wald’s test (Table 3).
$${\mathrm{Mb}}_{i,j,k,l}\;=\;\theta +\mu_{j}+\gamma_{k}+{\mathrm{dolphin}}_{i}+e_{i,j,k,l}$$
where for the ith dolphin belonging to kth species, ${\mathrm{Mb}}_{i,j,k,l}$ denotes the myoglobin concentration in the jth muscle, θ is the overall mean, $\mu_{j}$ is the effect of the jth muscle (mastohumeralis was taken as reference muscle and thus, $\mu_{\mathrm{mastohumeralis}}\;=\;0$) and $\gamma_{k}$ is the effect of the kth species ($\gamma_{D.\mathrm{delphis}\;}\;=\;0$ for *D. delphis*, which acted as the reference specie), ${\mathrm{dolphin}}_{i}$ is the random effect corresponding to i^th^ dolphin and $e_{i,j,k,l}$ is the variability within that cetacean (where the subscript l refers to the lth replicate effect). We assume that the random variables ${\mathrm{dolphin}}_{i}$ are independent and identically distributed $N\left(0;\sigma_{b}\right)$. Likewise, the errors ($e_{i,j,k,l}$) are also assumed to be independent and identically distributed $N\left(0;\sigma_{b}\right)$. The goodness of fit was evaluated using the likelihood-ratio test (LRT), Akaike information criteria (AIC) and Bayesian information criteria (BIC). The fixed effects were summarized as estimations, standard errors (SE) and P-values (Table 3). From this model, the Mb concentration’s values were estimated for each of the muscles using 95% confidence intervals.

Because CET 829 was the only juvenile in the sample, and Mb concentration changes across development [20,39], the mixed model was carried out both with and without this individual.

#### 2.4.2. Mb Concentration in Locomotor and Non-Locomotor Muscles

To compare the Mb concentration between the locomotor and non-locomotor muscles, the difference between the means of effects was considered:$$\begin{array}{c} \partial \;=\;(1/6)\;(\mu_{E.\mathrm{axilla}}+\mu_{E.\mathrm{middle}}+\mu_{E.\;\mathrm{anus}}+\mu_{H.\;\mathrm{middle}}+{\;\mu }_{H.\;\mathrm{anus}}+\mu_{R.\;\mathrm{abdominis}})\\ -(1/3)\;\left(\mu_{\mathrm{Mastohumeralis}}+\mu_{D.\mathrm{scalenus}}+\mu_{\mathrm{Sternohyoideus}}\right) \end{array}$$
namely: the mean of the Mb concentration corresponding to the locomotor muscles minus the mean of the Mb concentration of the non-locomotor muscles, and general hypothesis tests were applied.

#### 2.4.3. Mb Concentration and Muscle Mass Relationship

To test if there was a relationship between Mb concentration and muscle mass, the following mixed model was used:$${\mathrm{Mb}}_{i,j}\;=\;\mu \;+{\mathrm{dolphin}}_{i}+\beta \cdot \log\left({\mathrm{Muscle}\;\mathrm{Mass}}_{i,j}\right)+e_{i,j}$$
where ${\mathrm{Mb}}_{i,j}$ denotes the Mb concentration of jth muscle corresponding to ith dolphin, muscle Mass_i,j_ the muscle mass of the jth muscle and e_i,j_ is the variability within the dolphin, log stands for natural logarithm. The goodness of fit was summarized as the marginal R^2^ (represents the variance explained by the fixed effects) and the conditional R^2^ (is interpreted as a variance explained by the entire model, including both fixed and random effects). Muscle mass was logarithmically transformed to achieve a better fit.

#### 2.4.4. Comparison of the Functional Muscle Group Method with the Locomotor and Total Muscle Mass Methods

To compare the results from this functional muscle method with the more commonly used total and locomotor muscle methods for muscle O_2_ stored determination, we used the intraclass correlation coefficient (ICC) and represented the data graphically using scatterplots. The ICC between two numerical variables measures the degree of coincidence between the variable values.

For statistical analysis, the R package, version 3.3.1 [40], was used with statistical significance at *p* < 0.05. Mb concentration variation between species and muscle groups and locations was assessed using a mixed model (see methods 2.4.1), whereas the rest of the results are raw data. Dot and box plots were made using SPSS 17 (SPSS Inc., Chicago, IL, USA).

## 3. Results

The results shown below for Mb concentration variation between species and muscle groups and locations were assessed using a mixed model (see Section 2.4.1), whereas the rest of the results are raw data. In the Appendix A section, all data displayed are raw data.

### 3.1. Mb Concentration between and within Body Muscles

Across all species, the adjusted mean Mb concentration in the locomotor muscles was 2.62 g Mb · 100g^−1^ muscle (95% CI = 2.54–2.71) higher than in the non-locomotor muscles. Moreover, Mb concentration in locomotor muscles was significantly higher than in non-locomotor muscles (4.58 g Mb · 100g^−1^ muscle vs. 1.96 g Mb · 100g^−1^ muscle, *p* < 0.001). Heterogeneity in Mb concentration was observed among and within the various muscle groups studied (Table 4, Figure 2).

Mb concentration and muscle mass showed a strong positive relationship (Conditional R^2^ = 0.77), meaning that locomotor muscles had higher mass and higher Mb concentrations than non-locomotor muscles.

### 3.2. Mb Concentration among Species

Significant differences were observed in the overall Mb concentration (*p* ≤ 0.004 for *D. delphis* vs. *S. frontalis* and *p* < 0.001 for all other comparisons between species) among the three species included in the study. The highest Mb concentration was observed in *S. coeruleoalba*, followed by *D. delphis* and *S. frontalis* (Figure 3, raw data in Appendix A). 

### 3.3. Comparison between Reynafarje Equation and the Calibration Curve for Mb Concentration Determination

Mb concentration values were different when calculated using the Reynafarje equation and the calibration curve in dolphin CET 834 (Appendix A). Mean Mb concentration values were higher for the different muscles, except for the *sternohyoideus*, when calculated by the calibration curve method than with the Reynafarje equation. Although differences in Mb concentration values between the methods were subtler for non-locomotor muscles than for locomotor muscles, the calibration curve gave Mb concentration values over 1 g Mb · 100 g^−1^ muscle, higher than the values calculated using the Reynafarje equation.

### 3.4. Muscle O_2_ Storage Determination

The data presented in the current study, which evaluated Mb concentration heterogeneity across different functional groups, showed that the locomotor muscle groups were the major contributors to the total O_2_ storage (mean 92.3% vs. 7.7% for non-locomotor muscles). Among the different locomotor muscle groups, following the classification proposed in this study, the epaxial upstroke group stored the largest amount of O_2_ (mean 54.0% of total), followed by the hypaxial (mean 26%) and the *rectus abdominis* downstroke groups (mean 12.8%) (Figure 4, Appendix A). Interindividual differences among oxygen stored within the different functional muscle groups were also observed.

### 3.5. Comparison of the Functional Muscle Group Method with the Locomotor and Total Muscle Mass Methods

The locomotor muscle mass method underestimated total muscle O_2_ storage of cetaceans compared with the functional muscle group method by an average of 6.2%, as the O_2_ stored in non-locomotor muscles is not taken into account. In contrast, the total muscle mass method overestimated total muscle O_2_ storage when compared to the functional muscle group method by 13.0% since it relied upon a single Mb concentration value taken at the middle of the epaxial muscle, which is one of the highest measured in any muscle (see Table 4). Nevertheless, both previous methods showed a high intraclass correlation when compared with the functional muscle group method (ICC > 0.96 in both cases) (Figure 5).

## 4. Discussion

As shown in previous studies [19,20,21,22,23,24], myoglobin was heterogeneously distributed between and within muscles. In this study, statistically significant differences in Mb concentration were found between locomotor and non-locomotor muscles (Table 4). Locomotor muscles were the major contributors to total muscle O_2_ stores due to high Mb concentration and large muscle mass. Previous methods assuming homogenous Mb concentration distribution slightly underestimated muscle O_2_ stores when only locomotor muscles were considered, and overestimated it when total muscle mass was considered (when only one relatively high Mb concentration value was used to estimate oxygen).

### 4.1. Differences in Mb Concentration between and within Body Muscles

Locomotor muscles contained significantly higher Mb concentration than non-locomotor muscles in the three species studied (2.62 g Mb · 100 g^−1^ muscle higher than non-locomotor), which can be interpreted within the context of the functional roles of these muscle groups. Locomotor muscles generate the thrust to produce the tail fluke movement and are likely more metabolically active during a dive than non-locomotor muscles [30,31]. The cranial region of the epaxial muscle (axilla section) had a lower Mb concentration than the caudal regions. This result is consistent with previous observations that the cranial region of dorsal muscles contributes less to thrust generation than caudal regions [19,22,41]. However, previous studies described an increasing gradient in Mb concentration from the cranial to the caudal direction [19,22]. In contrast, in the present study, the highest Mb concentration value was found at the middle region, in agreement with the study carried out by Harrison and Davis [42], which suggests that species-specific patterns of Mb concentration exist.

Muscle mass was significantly positively correlated with Mb concentration in our study, meaning that larger muscles had also a higher Mb concentration per unit of muscle mass. Different factors have been proposed, at a molecular level, as drivers in Mb stores development during ontogeny, exercise being one of them [43]. Higher levels of Mb concentration in larger muscles, like locomotor muscles, may be explained due to high levels of exertion when compared smaller muscles, like non-lomotor ones. In relation to this, future studies on fiber size and fiber type (slow vs. fast fibers) should also be performed for different body muscles, including locomotor and non-locomotor, as they may provide important information to explain a higher or lower Mb concentration depending on the role of those muscles.

### 4.2. Mb Concentration among Species

The patterns of locomotor muscle and total muscle Mb concentration found in this study were consistent with those patterns displayed at the *longissimus dorsi* described by Kroeger et al. [27]. *S. frontalis* displayed the lowest Mb concentration among the three species. This species inhabits mostly shallower waters of the continental shelf and continental shelf break, and although they can perform dives between 40 and 60 m depth, most dives are less than 10 m [44]. *D. delphis* displayed a higher Mb concentration than did *S. frontalis* and this species is known to dive to at least 260 m to feed, but most dives do not exceed 100 m [45]. Finally, *S. coeruleoalba* showed the highest Mb concentration and O_2_ storage among the species studied. Although its foraging depth-range is reported to be between 200 and 700 m, it has been suggested that foraging occurs early at night when their prey is closer to the surface. Nevertheless, little is known about its diving behavior [46]. Deeper dives are generally longer, which requires larger O_2_ stores to complete the dives using aerobic metabolism, and likely explains this species’ higher Mb concentration.

Total body mass was not explored as a factor in the mixed model to estimate Mb concentration because the marine mammal literature indicates that Mb concentration is more related to the species and its ecophysiology (fast vs. slow swimmers or shallow vs. deep divers) than to the body mass itself. Data supporting this position are different Mb concentration values for species with similar body masses; small species possessing higher Mb concentration values than larger species (e.g., the species *P. phocoena*); or just the opposite, large-body mass species showing low Mb concentration values (e.g., baleen whales’ species), e.g., [34,47,48,49].

### 4.3. Comparison between Reynafarje Equation and the Calibration Curve for Mb Concentration Determination

Different Mb concentration values for the same muscle group within an individual were obtained using the Reynafarje equation and the calibration curve methods. Proper determination of total Mb in the body is essential to accurately estimate muscle O_2_ stores in the body and estimate ADL. Thus, we suggest using standard calibration curves, instead of Reynafarje’s equation, to allow a more reliable comparison of Mb concentration across laboratories and studies.

### 4.4. Muscle O_2_ Storage Determination

The locomotor muscle groups had higher calculated O_2_ stores than the non-locomotor muscles due to their larger muscle mass and higher Mb concentration. These results are similar to those of Dolar et al. [19], who described high muscle O_2_ stores in the locomotor muscles (epaxial and hypaxial groups and *rectus abdominis*) (82–86%) in Fraser’s dolphin (*Lagenodelphis hosei*), the spinner dolphin (*Stenella longirostris*), and the pygmy killer whale (*Feresa attenuata*). Large O_2_ stores within the locomotor muscle are essential because, as primary swimming muscles, they require a source of O_2_ to maintain aerobic muscle metabolism [9,15,22,34] (Appendix A). Differences observed in the O_2_ stored in the various functional groups among individuals from the same species may be explained due to interindividual differences. In the case of indivuals from the species *S. frontalis*, CET 822 showed the highest differences compared to the other three. This individual was a male compared to the other three that were females. Future studies, with a higher sample size, may be needed to determine if males of this species have more locomotor musculature than females.

### 4.5. Comparative Study between the Present and Previous Methods for Muscle O_2_ Stored Calculations

The locomotor muscle mass method underestimated dolphin adult muscle O_2_ stores compared to the functional muscle group method due to the omission of non-locomotor muscle mass, which had a mean muscle O_2_ stored percentage value of 7.7 ± 2.3% among the three species. In contrast, the total muscle mass method overestimated muscle O_2_ stores. This method extrapolates Mb values from the middle epaxial location, which has the highest Mb concentration, to all muscles.

Dolar et al. [19] determined Mb concentration and O_2_ stores in different muscles (epaxial, hypaxial, *rectus abdominis*, *intercostals*, *sternohyoideus*, *infraspinatus*, diaphragm and cutaneous muscles) of the cetacean species mentioned in Section 4.4. They obtained an overestimation, similar to our study, in muscle O_2_ stores when assuming homogeneous Mb concentration distribution using middle epaxial values. The differences in the mean overestimation values between both studies (7.6% vs. 13% of our study) may be due to both the choice of muscles and species-specific patterns of muscle Mb concentration heterogeneity.

The functional muscle group method presented in this study is labor-intensive, as it requires the measurement of Mb concentration in multiple muscles and dissecting and weighing most of the body muscles. Comparison with previous methods demonstrated a high intraclass correlation (Figure 5), meaning that both previous methods, which assumed homogeneous myoglobin concentration across the body’s skeletal muscles, accurately estimate total Mb and O_2_ stores. Nevertheless, in both methods, the locomotor and/or entire skeletal muscle mass should be weighed appropriately. Data of the relative muscle mass in the body for different species and age classes are scarce for marine mammals, e.g., [29,50,51,52,53]. Moreover, variation in body condition and muscle mass or differences among age classes and individuals need to be considered to influence the final myoglobin concentration e.g., [19,39]. Thus, individuals of different life history, reproductive, and nutritional classes must be studied to understand the onboard oxygen stores representative of the whole species fully.

## 5. Conclusions

As previous studies have demonstrated, Mb concentration was heterogeneously distributed between and within cetacean muscles investigated in this study. Locomotor muscles constituted the primary muscle O_2_ storage site due to their high Mb concentration and large muscle mass. This study demonstrates that using the locomotor muscle mass method yields quantitative data similar to those of the functional muscle group method, which is more time-consuming and technically challenging. Because locomotor muscles represent the overwhelming majority of the body’s skeletal muscle mass, this method provides accurate comparative insights into cetaceans’ diving and swimming capabilities whenever the locomotor muscles are properly dissected and weighed.

## Figures and Tables

**Figure 1 animals-11-00451-f001:**
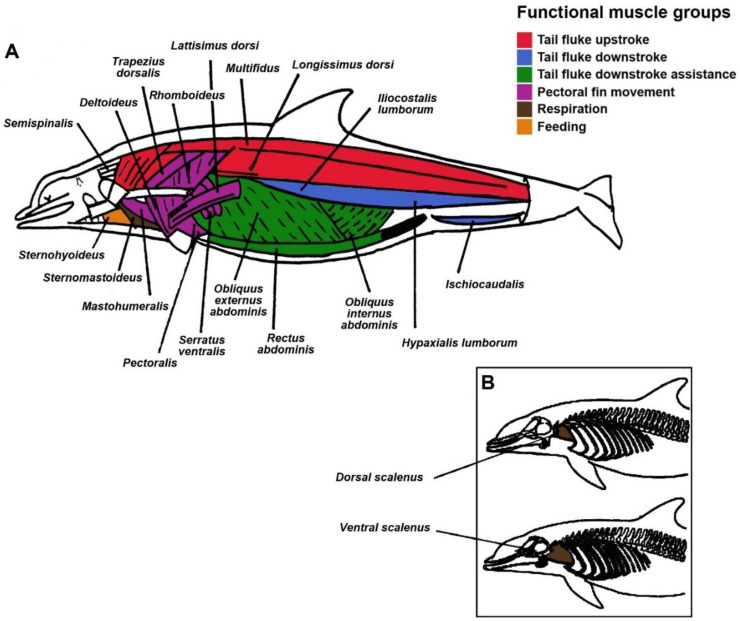
Dolphin body muscles grouped by colors depending on their function: (**A**) main locomotor muscles in red (epaxial; tail fluke upstroke), blue (hypaxial; tail fluke downstroke) and green (*rectus abdominis*; tail fluke downstroke); muscles involved in the pectoral fin movement in purple; muscles involved in feeding in orange; and finally, muscles involved in the respiration function in brown. Image modified from Rommel and Lowenstine [32]. (**B**) Dorsal and ventral scalenes in brown (involved in respiration). Image modified from Cotten et al. [31].

**Figure 2 animals-11-00451-f002:**
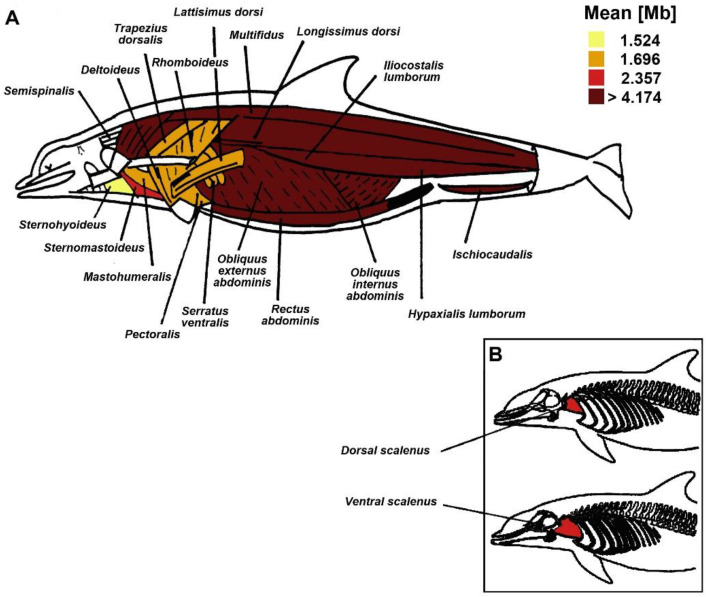
Dolphin functional muscle groups of the reference species (*D. delphis*). Muscles were colored based on the adjusted mean myoglobin (Mb) concentration value (g Mb · 100 g^−1^ muscle) of their representative muscles. Images modified from Rommel and Lowenstine [32] (**A**) and Cotten et al. [31] (**B**).

**Figure 3 animals-11-00451-f003:**
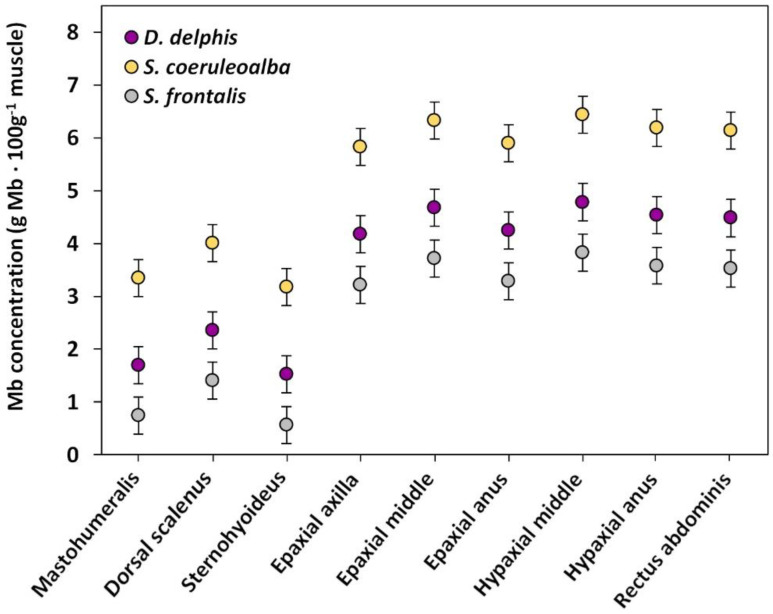
Mb concentration dot plot for the different muscles and muscle locations. Mb concentration was expressed by adjusted means (g Mb · 100 g^−1^ muscle) dot plot and 95% confidence intervals. The species were represented in different colors: *S. coeruleoalba* in yellow, *D. delphis* in purple and *S. frontalis* in grey.

**Figure 4 animals-11-00451-f004:**
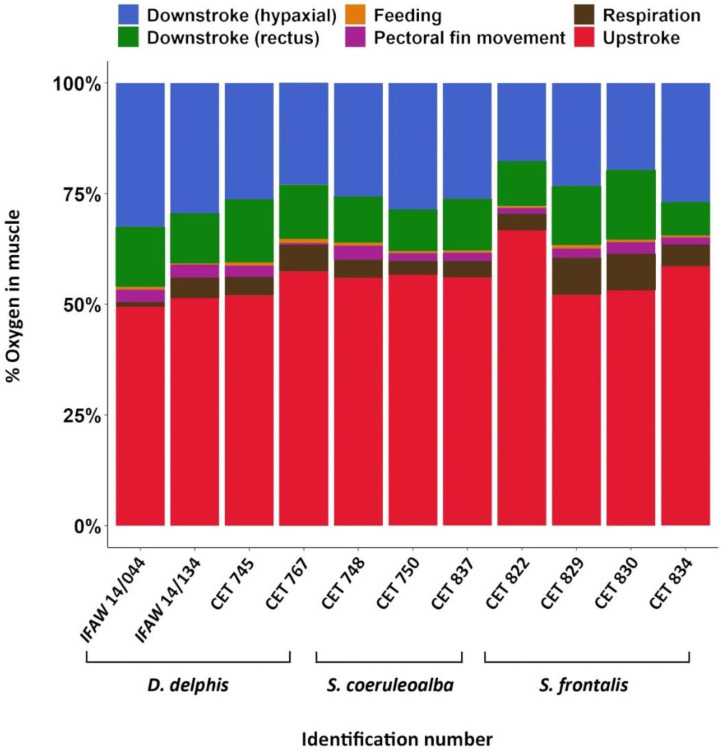
Percentage of O_2_ stored in the different functional groups considered of the animals studied. Individuals’ identification number is represented in the X-axis, and a different color representing each of the six functional groups considered. Raw data are presented here.

**Figure 5 animals-11-00451-f005:**
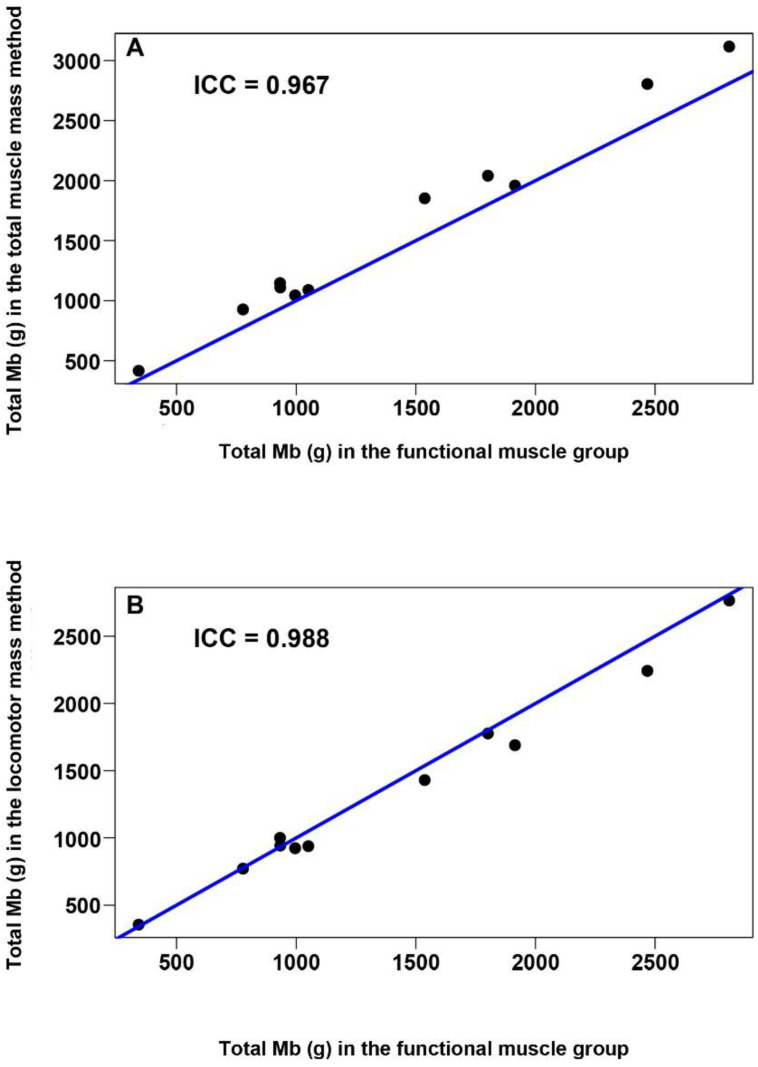
Total Mb comparison between the functional muscle method, proposed in this study, and previous methods (total and locomotor muscle mass methods). Scatter plots were represented comparing total Mb between our functional muscle group method and total muscle mass method (**A**) or locomotor muscle mass method (**B**). The blue line is the bisector, which would be the line resultant if the total Mb values, represented as dots, were coincidental for the two methods compared (intraclass correlation (ICC) = 1). In both graphs, the ICC was very close to 1, which means a high similarity between the total Mb calculated in the functional muscle group method and the locomotor (**A**) or total (**B**) muscle mass methods.

**Table 1 animals-11-00451-t001:** Biological and stranding conditions of the 11 animals included in the present study.

ID Number	Species	Age Class	Body Condition	Decomposition Code	Mass (kg)	Length (cm)	Sex	Stranding Location
IFAW 14/044	*D. delphis*	Subadult	Good	2	68.8	176	M	Cape Cod
IFAW 14/134	*D. delphis*	Subadult	Good	2	134	180	M	Cape Cod
CET 745	*D. delphis*	Adult	Poor	2	58.0	188	F	Canary Islands
CET 767	*D. delphis*	Adult	Poor	2	78.9	220	F	Canary Islands
CET 748	*S. coeruleoalba*	Subadult	Good	2	74.3	195	M	Canary Islands
CET 750	*S. coeruleoalba*	Adult	Poor	3	87.8	214	F	Canary Islands
CET 837	*S. coeruleoalba*	Subadult	Poor	3	70.0	188	M	Canary Islands
CET 822	*S. frontalis*	Adult	Moderate	2	61.4	170	M	Canary Islands
CET 829	*S. frontalis*	Juvenile	Moderate	2	30.3	137	F	Canary Islands
CET 830	*S. frontalis*	Adult	Moderate	3	59.8	172	F	Canary Islands
CET 834	*S. frontalis*	Adult	Moderate	3	65.4	175	F	Canary Islands

**Table 2 animals-11-00451-t002:** Classification of muscles based upon their primary function.

Locomotor Muscles			Non-Locomotor Muscles		
Head Movement	Tail Fluke Movement (Main Locomotor Muscles)		Pectoral Fin Movement	Feeding	Respiration
	Upstroke Movement	Downstroke Movement			
*Spinalis-semispinalis*	*Multifidus* *Longissimus dorsi* *Iliocostalis lumborum*	*Hypaxialis lumborum* *Ischiocaudalis* *Rectus abdominis* *Obliquus (internus and externus) abdominis*	*Mastohumeralis* *Deltoideus* *Levator scapulae* *Rhomboideus* *Lattissimus dorsi* *Serratus ventralis* *Pectoralis* *Trapezius*	*Sternohyoideus* *Sternothyroideus*	*Scalenus* *Intercostals* *Sternomastoideus*

**Table 3 animals-11-00451-t003:** Estimation of the fixed effects corresponding to the mixed model for myoglobin concentration (g Mb · 100g^−1^ muscle).

Fixed Effects	Coefficient (SE)	P (Wald)	P (LRT) *	AIC **	BIC ***
Intercept	1.696 (0.179)	<0.001			
Muscle			<0.001	3119.3	3143.2
*Mastohumeralis* (Ref.)	0	-			
*Dorsal Scalenus*	0.661 (0.087)	<0.001			
*Sternohyoideus*	−0.171 (0.089)	0.053			
Epaxial axilla	2.478 (0.088)	<0.001			
Epaxial middle	2.979 (0.087)	<0.001			
Epaxial anus	2.551 (0.087)	<0.001			
Hypaxial middle	3.086 (0.088)	<0.001			
Hypaxial anus	2.842 (0.087)	<0.001			
*Rectus abdominis*	2.786 (0.087)	<0.001			
Species			<0.001	1708.2	1760.8
*D. delphis* (Ref.)	0	-			
*S. coeruleoalba*	1.653 (0.258)	<0.001			
*S. frontalis*	−0.957 (0.239)	0.004			
*S. coeruleoalba–S. frontalis*		<0.001			

(*) Likelihood ratio test. (**) If the factor is deleted from the model. For the full model, Akaike information criteria (AIC) = 1683.3. (***) If the factor is deleted from the model. For the full model, Bayesian information criteria (BIC) = 1745.4. AIC and BIC are lack-of-fit measures. Note that when any factor is omitted from the model, the fit is worse. (Ref.) Reference category. (SE) Standard Error.

**Table 4 animals-11-00451-t004:** Myoglobin concentrations for muscles *D. delphis*: adjusted means by the mixed model. According to the mixed model, the adjusted means for *S. coeruleoalba* can be obtained, adding the quantity of 1.653, and for *S. frontalis* by subtracting 0.957. Raw data in Appendix A.

Functional Muscle Group	Muscle	Adjusted Means (95%CI)
Pectoral fin movement	*Mastohumeralis*	1.696 (1.346; 2.046)
Respiration	*Dorsal scalenus*	2.357 (2.006; 2.707)
Feeding	*Sternohyoideus*	1.524 (1.173; 1.875)
Tail fluke upstroke	Epaxial axilla	4.174 (3.823; 4.525)
	Epaxial middle	4.674 (4.324; 5.025)
	Epaxial anus	4.247 (3.897; 4.597)
Tail fluke downstroke	Hypaxial middle	4.782 (4.431; 5.133)
	Hypaxial anus	4.538 (4.188; 4.888)
Tail fluke downstroke assistance	*Rectus abdominis*	4.482 (4.131; 4.832)

## Data Availability

Data is contained within the article or Appendix A.

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
