# Peer review of "Myoglobin Concentration and Oxygen Stores in Different Functional Muscle Groups from Three Small Cetacean Species"

_animals, 2021, doi:10.3390/ani11020451_

Round 1

Reviewer 1 Report

Minor comments:

  1. The following change should be made throughout the manuscript
    1. [Mb] -> Mb
    2. at 28000 g for 50 min at 4°C -> at 28000 x g for 50 min at 4°C
  2. Some references cited in the manuscript should be updated to the recently published papers

Author Response

We want to thank you for revising the manuscript and the suggestions you have made to improve it. Below you can find the answer to your questions/suggestions.

- The following change should be made throughout the manuscript

[Mb] -> Mb

This change has been made throughout the manuscript. Now, Mb concentration is expressed as Mb concentration instead of [Mb].

-At 28000 g for 50 min at 4°C -> 28000 x g for 50 min at 4°C.

It has been changed (line 145).

-Some references cited in the manuscript should be updated to the recently published papers

Following your suggestion, we have added more recent references:

  1. Würsig, B.; Thewisen, J.G.M.; Kovacs, K.M. Encyclopedia of marine mammals; 3rd ed.; Academic Press, 2017; ISBN 9780128043271.
  2. Kooyman, G.L.; McDonald, B.I.; Williams, C.L.; Meir, J.U.; Ponganis, P.J. The aerobic dive limit: After 40 years, still rarely measured but commonly used. Comp. Biochem. Physiol. Part A 2021, 252, 110841, doi:10.1016/j.cbpa.2020.110841.
  3. Polasek, L.K.; Frost, C.; David, J.H.M.; Meyer, M.A.; Davis, R.W. Myoglobin distribution in the locomotory muscles of Cape fur seals (Arctocephalus pusillus pusillus). Aquat. Mamm. 2016, 42, 421–427, doi:10.1578/AM.42.4.2016.421.
  4. Davis, R.W. Physiological adaptations for breath-hold diving. In Marine mammals: Adaptations for an aquatic life; Springer International Publishing: New York, 2019; pp. 133–175 ISBN 978-3-319-98280-9.

Please let us know if there are still missing references.

-Moderate English changes should be performed in the manuscript.

English native coauthors have thoroughly reviewed English.

Attached you will find the reviewed manuscript. 

Thank you very much for your time and suggestions. 

Kind regards,

Reviewer 2 Report

Comments in document

Author Response

We would like to thank you for revising the manuscript and the suggestions you have made to improve it. Below you can find the answer to your questions/suggestions.

-The graphics of the dolphins are in poor resolution and appear to have been colored originally in pencil or crayon. While they are suitable and can be interpreted, I highly recommend making some improvements by using a photo editing program to color the internal sections of your models with a uniform color. The labels of each sampled region should be enlarged.

Both images (Figure 1 and 2) have been improved using a photo editing program. Moreover, labels have been enlarged to the maximum possible, considering that they are close among them. Additionally, we have added a legend that will make it easier for the reader to know at a simple glance the functional groups represented by the different colors.

-It might help the reader interpret the graphic and Table 2 O2 if each location was named and table was appended with numbers for each location. As is, it takes a bit of focus to relate the various names.

To make it easier for the readers, we have added a legend with the colors corresponding to the different muscle groups instead of the reviewer's suggestion as we thought the image could become confusing.

-Figure 3 – Make the x-axis diagonal. It strains the eyes to read it as is.

Following the reviewer's suggestion, the labels in the x-axis have been put diagonally.

-Figure 4 – Feeding is not clearly discernable. Perhaps change the coloring or add a border.

We have changed all the functional muscle groups' colors, making them coincidental with the colors previously used in figures 1 & 2. Thus, each color corresponds to the same functional group in all figures. We believe the change of colors makes it easier to discern the feeding group now.

-Figure 5 – Add parenthesis in x and y-axis labels to distinguish between axis title and values measured.

The x and y-axis have been relabelled, and Mb units (grams) were put within brackets.

-Figure 5 – First line of legend is confusing. State the different methods in the first line.

The initial sentence: "Total Mb comparison between our and other methods" has been changed by "Total Mb comparison between the functional muscle method, proposed in this study, and previous methods (total and locomotor muscle mass methods)" (lines 386 & 387).

-Line 320 – Reword "which would be the line", I think you mean to say it would represent the relationship or some other wording.

The sentence has been reworded to express that the bisector is the line resultant if the total Mb values, represented as dots, were coincidental for the two methods compared (ICC = 1) (lines 389-391).

-Line 95 – Sentence is long and gets confusing.

The initial sentence has been split into two sentences (lines 102-106).

-Line 162 and 167- Space after closing parenthesis.

Spaces have been added in both lines (lines 171 and 177)

-Line 242 – Should were be was?

Were has been changed by was (line 266).

We hope to have addressed properly all the changes/suggestions you have proposed. Thank you again for your time and suggestions, they have improved considerably the manuscript. 

Kind regards, 

Reviewer 3 Report

The manuscript entitled ‘Myoglobin concentration and oxygen stores in different functional muscle groups from three small cetacean species’ by Arregui et al. represents an interesting study where the authors evaluated the concentration of Myoglobin in different functional muscle groups as well as the oxygen storage. Three different cetacean species were evaluated namely S. coeruleoalba, D. delphis and S. frontalis. The results presented were very clear and indicated that [Mb] was heterogeneously distributed between and within cetacean muscles investigated in this study. Locomotor muscles constituted the main muscle O2 storage site due to their high [Mb] and large muscle mass.

In overall, I consider that the premise of this study is very interesting and important for the field and I will perform some comments and suggestions.

Major concerns:

  1. ‘Muscle mass was significantly positively correlated with [Mb] in our study’ (line 347). The authors should explore deeper this issue in discussion, proposing the underlying molecular mechanisms that should be addressed in future studies.
  2. Organize the Table 4 accordingly the functional groups used.
  3. In the Figure 2, the colours used are difficult to follow, specially the yellow (1,524) and the light orange (1,690).
  4. The analysis of the Figure 4. Percentage of O2 stored in the different functional groups considered of the animals studied divided into the 3 species. There is some heterogeneity in the percentage of O2 stored, according to the animal tested. These differences are higher in 2 of species analysed namely delphis and S. frontalis. The results were similar in the 3 animals tested for S. coeruleoalba. The authors should discuss these differences in the discussion section. Also, given that sample is very little they should provide an explanation for this point.

Author Response

We would like to thank you for revising the manuscript and the suggestions you have made to improve it. Below you can find the answer to your questions/suggestions.

'Muscle mass was significantly positively correlated with [Mb] in our study' (line 347). The authors should explore deeper this issue in discussion, proposing the underlying molecular mechanisms that should be addressed in future studies.

A paragraph detailing the molecular mechanisms that may explain a higher Mb concentration and futures studies that should be performed to address this has been added in lines 421-432.

-Organize Table 4 according to the functional groups used.

A new column with the functional groups to which each of the table's muscles belongs has been added.

-In Figure 2, the colors used are difficult to follow, especially the yellow (1,524) and the light orange (1,690).

Figure colors (Figures 1, 2 & 4) have been improved using a photo editing program. Following your suggestion, we have changed the colors of figure 2 and hope it is easier to distinguish groups now.

The analysis of Figure 4. Percentage of O2 stored in the different functional groups considered of the animals studied divided into the 3 species. There is some heterogeneity in the percentage of O2 stored, according to the animal tested. These differences are higher in 2 of species analyzed namely delphis and S. frontalis. The results were similar in the 3 animals tested for S. coeruleoalba. The authors should discuss these differences in the discussion section. Also, given that sample is very little they should provide an explanation for this point.

Following the reviewer's suggestion, a paragraph addressing the differences in oxygen stores in the various functional groups among individuals of the same species has been added between lines 472-478. Also, in the result section, a sentence highlighting these differences has been added (lines 350-352). Moreover, we have realized that we made a mistake in the value assigned to the respiration functional group for CET 767 when preparing the figure. We have changed it in the figure and the supplementary materials (Table S4).

We hope to have appropriately answered all your suggestions; otherwise, let us know. You will find attached to this answer the reviewed manuscript. 

Kind regards, 
